# Short communication: Runout of rock avalanches limited by basal friction but controlled by fragmentation

Øystein. T. Haug[1,2], Matthias Rosenau[1], Michael Rudolf[1], Karen Leever[1,3], and Onno Oncken[1]

[1]GFZ German Research Centre for Geosciences, Helmholtz Centre Potsdam, Telegrafenberg, 14473 Potsdam, Germany.
[2]Njord center, Department of Geosciences, University of Oslo, PO Box 1048, 0316 Oslo, Norway.
[3]Van Hall Larenstein University of Applied Sciences, Larensteinselaan 26a, 6882 Velp, The Netherlands.

**Correspondence:** Matthias Rosenau (rosen@gfz-potsdam.de)

**Abstract.** Rock avalanches produce exceptionally long runouts that correlate with their rock volume. This relationship has been attributed to the size-dependent dynamic lowering of the effective basal friction. However, it has also been observed that runouts of rock avalanches with similar volumes can span several orders of magnitude, suggesting additional controlling factors. Here, we analyze analog models of rock avalanches, with the experiments designed to test the role of dynamic fragmentation. We
show that for a fixed low basal friction, the runout of experimental rock avalanches varies over two orders of magnitude and is determined by their degree of fragmentation while the basal friction acts only as an upper limit on runout. We interpret the runout's dependence on fragmentation to be controlled by the competition between mobility enhancing spreading and energy-consuming fragmentation limited by basal friction. We formalize this competition into a scaling law based on energy conservation which shows that the variation in the degree of fragmentation can contribute to the large variation in runout of
rock avalanches seen in nature.

## 1   Introduction

With volumes larger than $10^9$ m$^3$, and speeds reported at over 150 km/h (Campbell, 1989) and possibly up to 100m/s (Legros, 2002), the destructive power of rock avalanches is unprecedented. They are exceptional hazards produced when very large
rockslides disintegrate during transport (Hungr et al., 2013). The travel distance of the deposit front, or runout, is an important measure for hazard assessment (Vaunat and Leroueil, 2002) and is generally found to be more than ten times longer than the fall height (Hsü, 1975). This suggests low effective basal friction $\mu_{eff}$, which is usually attributed to various dynamic weakening processes (e.g. Kent, 1966; Shreve, 1968; Hsü, 1975; Melosh, 1979; Campbell, 1989; Pudasaini and Miller, 2013; Legros, 2002; Lucas et al., 2014; Wang et al., 2017) or additional basal erosion processes (e.g. Hungr and Evans, 2004; Pudasaini and
Fischer, 2020).

Field observations of the displacement of rock avalanches are typically given by the ratio of vertical ($H$) and horizontal ($L$) distance from the deposit's front to the top of the main scarp. The resulting ratio

$$\mu_{apparent} = \frac{H}{L} \tag{1}$$

is known as the Heim's ratio (Heim, 1882, as cited in Hsü, 1975) and serves as a proxy for $\mu_{eff}$ when called the "apparent" coefficient of friction (Manzella and Labiouse, 2012). One of the best established, but perhaps least understood observations

of rock avalanches, is the dependence of the Heim's ratio on volume: rockslides below a size of approximately $10^6 \, \mathrm{m}^3$ all have a relatively constant Heim's ratio of ∼0.4-0.7, but for larger rockslides Heim's ratio decreases with volume, reaching values <0.1 for volumes larger than $10^9 \, \mathrm{m}^3$ (Pudasaini and Miller, 2013; Lucas et al., 2014). This suggests a scale-dependent mechanism of decreasing apparent friction with volume that becomes dominant at large volumes (Davies and McSaveney, 1999). Analytical modeling and numerical simulation involving lubrication mechanisms by Pudasaini and Miller (2013) and

Lucas et al. (2014) provided mechanical explanations for this observation. Importantly, however, even within a narrow range of volumes, runouts are seen to span orders of magnitude suggesting additional controlling factors on runout that are insensitive to size. Runout variability lies in part also in the fact that the runout is defined by the front of the deposits, and therefore contains the combined effect of both translation and spreading of the rock mass. The additional travel distance caused by spreading can have a profound effect on the runout (Staron and Lajeunesse, 2009), especially if the effective basal friction is low.

Recently, the process of dynamic fragmentation has received increased attention from the reserach community, and much progress has been made in our understanding of its role in the dynamics of rock avalanches (Locat et al., 2006; Imre et al., 2010; Bowman et al., 2012; Pudasaini and Miller, 2013; De Blasio and Crosta, 2015; Haug et al., 2016; Zhao et al., 2017, 2018; Lin et al., 2020; Gao et al., 2020; Knapp and Krautblatter, 2020). Firstly, one may expect that the finer the material, the more flow-like the behavior, increasing its mobility and allowing the rock mass to spread more easily (Locat et al., 2006; Wang et al.,

2017; Zhao et al., 2018). Secondly, models of fragmenting rockslides suggest that dynamic fragmentation actively increases the spreading (Bowman et al., 2012; De Blasio and Crosta, 2015; Lin et al., 2020). However, fragmentation has also been shown to consume energy (Haug et al., 2016; Zhao et al., 2017; Lin et al., 2020), potentially at a cost to the runout length. Clearly, understanding the integrated effect of fragmentation on the runout dynamics of rock avalanches requires more analysis.

To study the effects of friction and fragmentation on rock avalanche dynamics, we here analyze analog models of dynami-

cally fragmenting rock slides. We assume that there exists some mechanism (or a set of mechanisms) that causes a low effective coefficient of basal friction which we set to 0.15-0.20 in our models. To isolate the scale-independent effect of fragmentation we keep both the volume and friction within a narrow range in our models compared to nature. Note, this approach explicitly excludes dynamic weakening mechanisms that are suspected in natural prototypes. Specifically, our models do not include fluids and frictional heating is insignificant such that lubrication mechanisms (e.g. Pudasaini and Miller, 2013; Lucas et al.,

2014) do not play a role. Granular pressurization (e.g. Imre et al., 2010) is also not considered significant in our experiments because of the low energy involved. Other potentially important mechanisms like bedrock erosion (e.g. Hungr and Evans, 2004; Pudasaini and Fischer, 2020) are excluded here for simplicity. The experimental design, therefore, means that the observed variation in Heim's ratio is due to fragmentation and dry friction. We describe the dependence observed between the

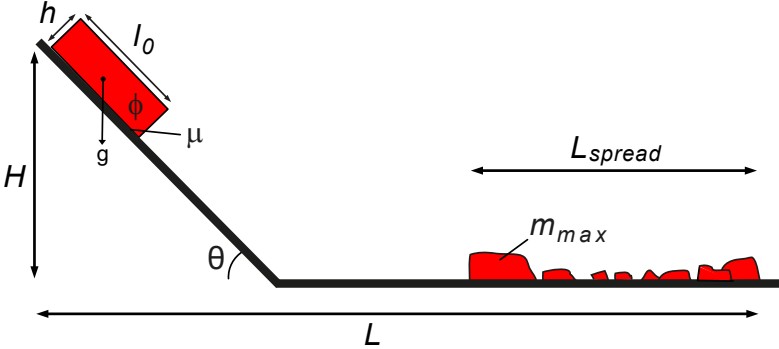

**Figure 1. Sketch of the slope geometry of experiments, relevant parameters, and length scales (modified after Haug et al. (2016))**

runout and the degree of fragmentation in the form of a scaling law. Finally, we compare our experimental results to a set of
natural data and discuss their relevance to natural systems. All data underlying this study as well as additional relevant data are
published open access in Haug et al. (2020).

## 2 Experimental methods

In the experiments, originally documented in Haug et al. (2016), a block of height $h$ and length $l_0$ (width = $l_0$) of rock analog
material is gravitationally accelerated down a plate held at an angle of $45°$ to the horizontal (Figure 1). After 1 m of travel,
the sample impacts a horizontal plate causing it to fragment. Once the sample fragments have slid onto the horizontal plane,
they spread and decelerate due to the internal and basal frictional interaction, before finally coming to rest. We use silicate
glass as our substrate, on which the basal friction coefficient is ca. 0.15-0.20 (Haug et al., 2016) - similar to the lowermost
values found in natural prototypes (Pudasaini and Miller, 2013; Lucas et al., 2014). The analog rock material is a cemented
fluvial quartz sand. The sand is cemented by mixing it with water and gypsum or potato starch and is left to set for 2 days
(for gypsum cement) or heated for 15 minutes in a 900 W microwave (for potato starch cement). The cohesion of the material
can be controlled by the type and amount of cement added to the mixture, allowing control of the strength of the material over
several orders of magnitude. The internal friction coefficient relevant for fragmenting intact material is 0.7 and reduces to 0.6
when fragments interact (see Haug et al., 2014, 2016, for details on the experimental setup).

The three main observables from the experiments are: (i) the degree of fragmentation ($m_c$), (ii) the Heim's ratio ($H/L$), and
(iii) the normalized deposit length ($L_{spread}/H$). We characterize the degree of fragmentation through the total mass of the
sample divided by the mass of the largest fragment ($m_c = M/m_{max}$). We choose this rather simple parameter, which has been
validated and benchmarked against breakage parameters used by previous studies in Haug et al. (2016), as a tradeoff between
capturing the process accurately in models and the accessibility of the equivalent information in nature. To define $L_{spread}$, we
consider the mass-weighted average position of the most proximal and distal 5 % of the total mass. This defines a rim which
is a more robust runout estimate than using single fragment positions as used by Haug et al. (2016) and is at the same time

accessible both experimentally and empirically. We normalize $L_{spread}$ by fall height $H$ in order to have a parameter describing the conversion of potential energy into spreading equivalent to Heim's ratio.

The experimental data analyzed here are coming from two series of experiments with varying degree of fragmentation: (i) one series of experiments where the thickness to length ratio ($h/l_0$) of the samples has been varied between 0.033 and 0.49 (corresponding to a one order of magnitude range in volume) while keeping the cohesion ($C$) constant at 14 kPa. (ii) one series of experiments where the cohesion of the material is varied between 4 and 350 kPa while keeping the thickness to length ratio constant at 0.13. In both series of experiments, the fall height ($H$) is kept constant at 0.71 m. Interested readers are referred to Haug et al. (2016) for details on the effect of cohesion and geometry on the degree of fragmentation. Additionally, two new experiments were performed to study the moment of fragmentation at high temporal resolution. For these experiments, the fragmentation of two samples with different cohesions but equal geometry ($C = 4$ and 40 kPa, $h/l_0 = 0.13$) is considered. These two experiments have a fall height of 0.35 m, and data is captured by a camera with a frame rate of 500 Hz (see Haug et al., 2020, for movies of these experiments). Combining these sets of data from various experiments allows for covering a wide enough parameter space for the analysis in this study.

## 3 Results and discussion

### 3.1 Experimental observations and interpretation

Figure 2 presents snapshots from two representative experiments, one with an intermediate strong sample and one with a low strength sample, illustrating the process of fragmentation. The stronger sample (Figure 2A) is observed to fragment less than the weaker one (Figure 2B). Thereafter, fragments of the stronger sample spread with limited interaction while the fragments from the weaker sample collide and/or slide next to each other and deposition starts relatively early. We infer, at first order, that while mobility generally increases with fragmentation, a higher amount of internal deformation is experienced along with increased fragmentation and increased deposition.

To quantitatively analyze the experiments, we focus on the correlation between runout and fragmentation and neglect all other parameters. This is justified by the collapse of experimental and natural data when plotting Heim's ratio against fragmentation in Figure 3A. Qualitatively, Heim's ratio decreases rapidly for low to intermediate degrees of fragmentation, reaching a minimum at $m_c \approx 5$ of about 0.2 and increases again slightly for higher degrees of fragmentation. A similar relation is observed between the length of the deposits (Figure 3B), which increases with fragmentation until $m_c \approx 5$ and slightly decreases beyond.

The rapidly decreasing Heim's ratio for $m_c < 5$ observed in Figure 3A is likely linked to the increased spreading with fragmentation seen in Figure 3B. A similar result was also obtained by previous analog experiments (Bowman et al., 2012; Haug et al., 2016) as well as numerical models (De Blasio and Crosta, 2015; Zhao et al., 2017). However, here we show that the Heim's ratio is not simply decreasing with the degree of fragmentation, but that it displays an optimum for $m_c \approx 5$. Importantly, the lowest apparent basal friction, equivalent to the lowest Heim's ratio, is close to the implemented basal friction (i.e. friction coefficient of 0.15-0.20 between samples and glass). Therefore, all processes operating in our models (e.g. fragmentation,

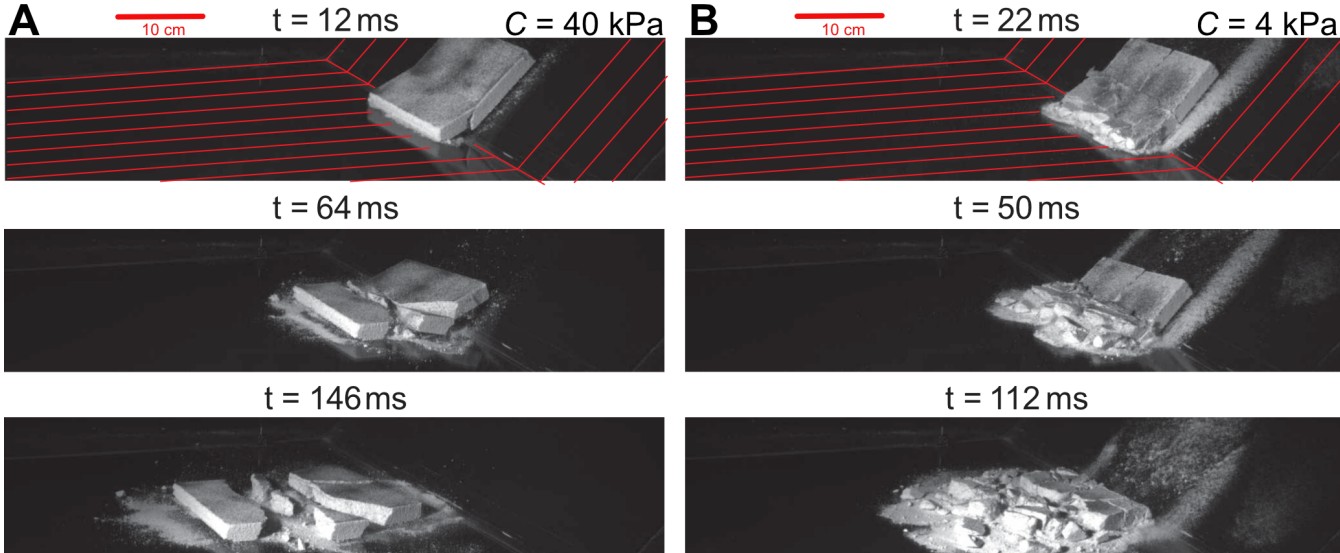

**Figure 2. Snapshots from the experiments**: (A) intermediate strength sample ($C = 40\,\mathrm{kPa}$) and (B) low strength sample ($C = 4\,\mathrm{kPa}$). The red lines in the upper images indicate the geometry of the basal plates. Images are chosen to represent similar travel distances in (A) and (B). The time given above each image reflects the time since the first impact. The samples have dimensions 15x15x2 cm. Note that the stronger sample (A) breaks apart into six large fragments with a limited amount of fine material produced and moves apart with little interaction after breaking. In contrast, the weaker sample (B) fragments into many small pieces with a large fraction of fine material causing frictional interaction and that deposits relatively early. Movies of the experiments are available in Haug et al. (2020).

internal friction between fragments, deposition) tend to consume energy and thereby reduce runout from its optimum (Haug
et al., 2016). Considering the increased internal deformation observed with the degree of fragmentation (Figure 2), the reduction of runout and spreading for $m_c > 5$ appears to be the result of the increased energy dissipation through internal friction within the rock mass as well as an increase in basal friction as the sliding surface becomes rougher due to syn-sliding deposition (e.g. Pudasaini and Fischer, 2020). A loss of mass and therefore momentum due to deposition may additionally result in deceleration and reduced runout as a function of $m_c$ (e.g. Pudasaini and Fischer, 2020). Consequently, the minimum of the
Heim's ratio observed in Figure 3A appears as the result of a competition between the spreading enhancing mobility and the energy-consuming fragmentation process.

## 3.2   A scaling law for runout

The interplay between fragmentation and friction in a dry environment can be formalized into a scaling law by considering the conservation of energy. Generally, the conservation of energy of a sliding mass $M$ requires that

$$MgH = \mu MgL_p + W \tag{2}$$

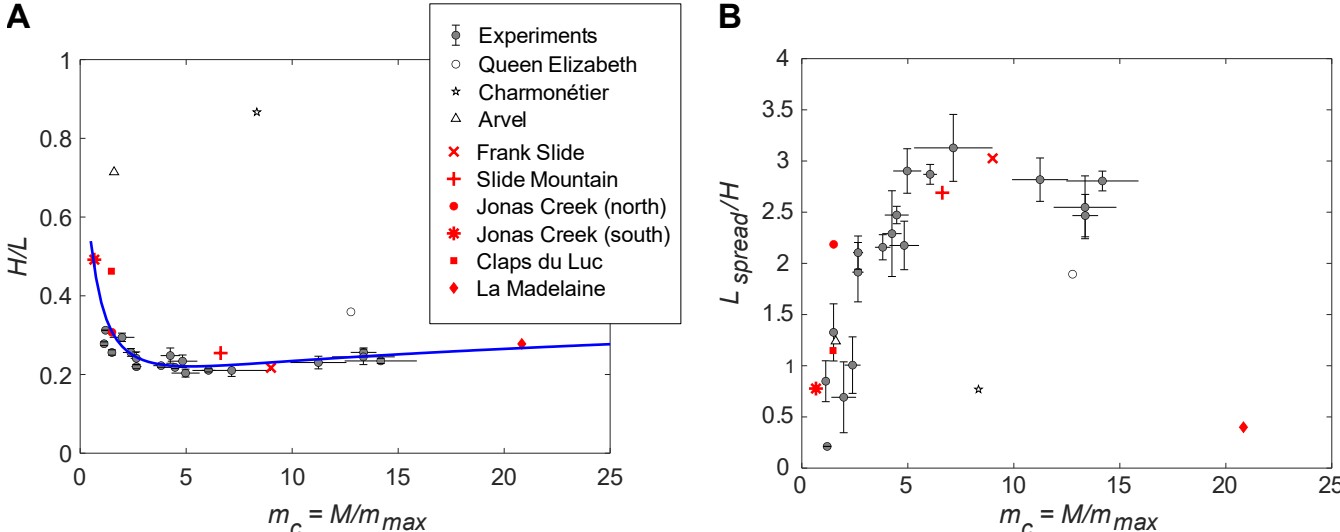

**Figure 3. Heim's ratio and deposit length of experiments (this study) and natural rock avalanches (from Locat et al., 2006).** (A) The Heim's ratio of the analog experiments (grey) and from the rock avalanches (red = selected set, open = discarded). The blue line represents the best fit of Equation 8 to experimental and natural data with parameters $\alpha = -1.0$, $\beta = 1.5$, $\gamma = 0.11$. (B) The deposit's lengths. In both panels, the grey circles represent the average value of a set of 4-15 experiments and the error bars give the standard error of the set. Note the opposite trends of the two curves suggesting an intrinsic relationship between spreading and runout.

where $g$ is the gravitational acceleration, $H$ is the vertical fall height, $L_p$ is the entire travel path of the slide and $W$ is the sum of any other energy dissipating terms. Here, we have assumed a Coulomb friction coefficient $\mu$ at the base.

For the geometry of our experimental setup (see Figure A1), and also roughly for the set of selected rock avalanches, the $L_p$ can be expressed in terms of the horizontal runout $L$ as

$$L_p = L + L_s(1 - \cos\theta) - \frac{1}{2}l_0 - \frac{1}{2}L_{spread} \tag{3}$$

where $L_s$ is the length and $\theta$ the angle of the slope, and $l_0$ is the initial length of the slide. It is assumed that the additional travel length due to spreading is equal to half the deposit length ($L_{spread}$). Since $l_0$ is expected to be very small compared to the other terms, it is neglected in further analysis. Inserting Equation 3 into Equation 2 and solving for $L$ gives

$$L = \frac{H}{\mu} - L_s(1 - \cos\theta) + \frac{1}{2}L_{spread} - \frac{1}{\mu Mg}W \tag{4}$$

where it is emphasized that both $L_{spread}$ and $W$ are expected to be functions of the basal friction, $\mu$, internal friction, $\phi$, the degree of fragmentation, $m_c$, as well as a possible non-linear dependence between $L_{spread}$ and $W$. Rearranging Equation 4 yields the Heim's ratio in the form of

$$\frac{H}{L} = \mu\left(1 - \frac{\mu}{\sin\theta}(1 - \cos\theta) + \frac{\mu}{2H}L_{spread} - \frac{1}{MgH}W\right)^{-1}. \tag{5}$$

A direct determination of the two last terms in Equation 5 is difficult. However, based on the shape of the function of both the Heim's ratio and the $L_{spread}$ plotted in Figure 3, it appears that it can be reasonably described by an exponential function of $m_c$:

$$\frac{\mu}{2H}L_{spread} = \alpha e^{-m_c/\beta}. \tag{6}$$

Additionally, the experimental work by Haug et al. (2016) suggests that dissipative energy loss through fragmentation increases less for higher degrees of fragmentation and therefore can be described with a logarithmic function of $m_c$:

$$\frac{1}{MgH}W = \gamma log(m_c). \tag{7}$$

Using these approximations, Heim's ratio can be expressed as

$$\frac{H}{L} = \mu \left(1 - \frac{\mu}{\sin\theta}\left(1 - \cos\theta\right) + \alpha e^{-m_c/\beta} - \gamma log(m_c)\right)^{-1} \tag{8}$$

where $\alpha$, $\beta$, and $\gamma$ are constants to be empirically determined.

This equation describes the competition between spreading (proportional to $e^{-m_c/\beta}$) and the increasing energy dissipation (proportional to $log(m_c)$) with $m_c$ and its relation to friction. A best fit of this function to the natural and experimental data is presented in Figure 3A (blue line), where $\alpha = -1.0$, $\beta = 1.5$, $\gamma = 0.11$. A fit constrained only the experimental data yields

very similar results ($\alpha = -0.68$, $\beta = 2.0$, $\gamma = 0.11$, see Figure B1). This suggests spreading dominates runout for low degrees of fragmentation (i.e. $m_c < 5$), but has little effect at high degrees of fragmentation as the exponential term approaches zero. At high degrees of fragmentation, the energy dissipation related to fragmentation, therefore, becomes increasingly relevant in controlling runout. At $m_c \approx 5$, i.e. when about 80-85 % of the volume is fragmented, a state of optimal mobility is reached with a Heim's ratio limited by the basal friction coefficient suggesting that energy is consumed mainly by basal friction, which

then is the limiting factor for runout.

## 3.3   Application to a natural data set

We compare our experimental results in Figure 3 with data from nine rock avalanches reported by Locat et al. (2006) that show no clear volume dependence of runout. This feature makes this data set ideal for testing whether a scale-independent process is operating besides dynamic basal weakening. However, not all the rock avalanches reported in (Locat et al., 2006)

are comparable to our experimental setup concerning material properties and geometries (Figure 1). Based on slope geometry, the Queen Elizabeth slide is discarded because of its run-up on the opposite valley wall. Also discarded is the Charmonétier slide because of the sudden free-fall stage at the end of its transport. Additionally, the Arvel slide was observed to bulldoze soft material in front of it, and such complexities are not considered in our models so this one is also neglected. Note that in all three discarded cases, the late-stage processes tend to increase the expected Heim's ratio and they consistently plot above the

trend of the other data in Figure 3B.

Figure 3 displays remarkably similar trends between the experimental and the selected natural data that all follow the proposed scaling law. The data points from Jonas Creek (north) and Clapse du Luc are observed to extend the trend from

the experiments to higher Heim's ratios for low degrees of fragmentations while La Madelaine slide is observed to extend the trend of the experimental results of Heim's ratio to higher degrees of fragmentation (Figure 3A). Its low spreading value (Figure 3B) suggests that the reduction of spreading indicated by the experiments for $m_c > 5$ continues for even higher degrees of fragmentation. The agreement between these slide deposit lengths and the extrapolation of the experimental trend through Equation 8 (Figure 3B) supports the validity of our proposed scaling law. The Heim's ratios of the neglected slides are all, as expected, higher than the selected data set for their respective degrees of fragmentation, illustrating the importance of topography (e.g. opposite valley wall) and processes such as bulldozing.

The similarity seen between experimental and natural data suggests some universality concerning the empirical constants. Moreover, the similarity suggests that the rock avalanches considered here all have a close to constant effective basal friction of about 0.15-0.20. This implies that over the range of two orders of magnitude (from $2 \cdot 10^6$ to $90 \cdot 10^6$ m$^3$) represented by this data set, the effective coefficient of friction of rock avalanches could be considered independent of volume. Consequently, our results suggest that the variation seen in Heim's ratio for these rock avalanches is not (only) caused by scale-dependent basal friction, but by differing degrees of fragmentation. This shows that fragmentation plays a governing role in the runout of rock avalanches and should be included in hazard assessments.

## 4   Conclusions

We studied the dynamics of fragmenting rock avalanches experimentally to unravel the control of basal friction versus fragmentation on runout behavior. We find that fragmentation causes both spreading and frictional interaction - competing processes that control the avalanche dynamics. Based on energy arguments we derive a scaling law with empirical constants that quantifies the relative importance of spreading and frictional interaction as a function of fragmentation. The scaling law approaches an extreme for which runout is maximized and limited only by basal friction, which itself might be volume-dependent as suggested by earlier studies. The scaling law is validated against a natural data set verifying its applicability.

*Data availability.*  The data for this paper is available as an open access data publication (Haug et al., 2020).

*Video supplement.*  Videos for this paper is available as an open access data publication (Haug et al., 2020).

*Author contributions.*  OTH designed and run the experiments, derived the scaling law and wrote the first draft of the manuscript. MRo and MRu assisted in the experiments. MRo, KL and OO were involved in study design. All authors contributed to discussion and writing.

*Competing interests.*  No competing interests are present.

*Acknowledgements.* The authors would like to thank to Frank Neumann and Thomas Ziegenhagen for construction and technical assistance.

The work has been supported by the Helmholtz Graduate Research School GEOSIM, the German Ministry for Education and Research (BMBF, FKZ03G0809A) and the Deutsche Forschungsgemeinschaft (DFG) through grant CRC 1114 "Scaling Cascades in Complex Systems" (No. 235221301) project B01. We thank Kirsten Elger and GFZ Data Services for publishing the data. We thank the editor Michael Krautblatter and the two anonymous reviewers which provided very helpful comments which improved the manuscript. We thank Jon Bedford for providing a thourough language check of the manuscript.

## Appendix A: Definitions

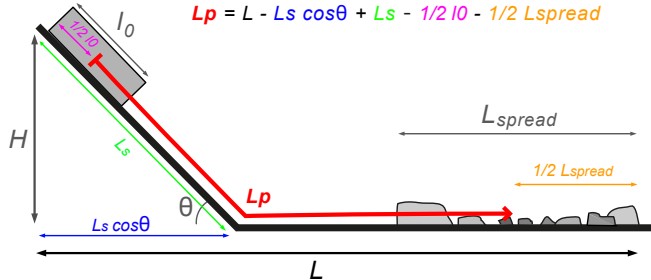

**Figure A1. Definition of distances used in Equation 3** . $L_p$ is the length of the travel path.

## Appendix B: Scaling law fit to experimental data only

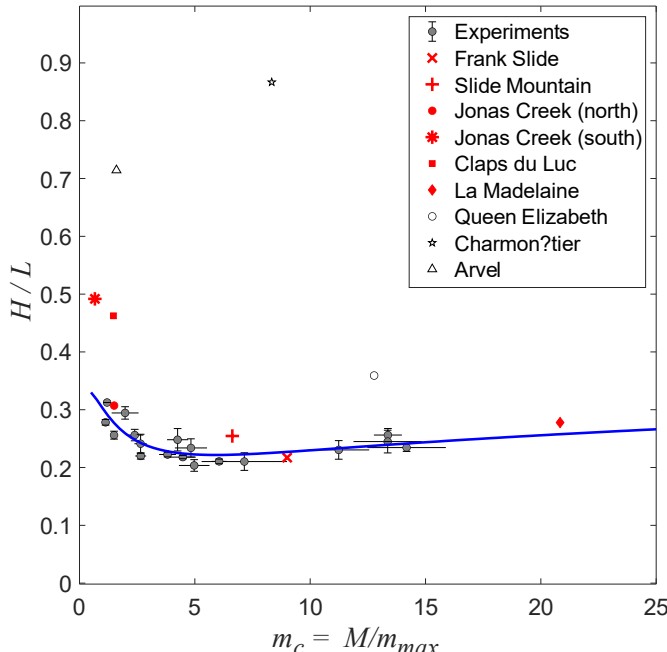

**Figure B1. Heim's ratio and deposit length of experiments (this study) and natural rock avalanches (from Locat et al., 2006)**. The Heim's ratio of the analog experiments (gray) and from the rock avalanches (red = selected set, open = discarded). The blue line represents the best fit of Equation 8 to experimental data with parameters $\alpha = 0.11$, $\beta = 0.68$ and $\gamma = 2.0$. Data shown and a Matlab-script to plot them are available in Haug et al. (2020).

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
