# Peer review of "Short communication: Runout of rock avalanches limited by basal friction but controlled by fragmentation"

_Earth Surface Dynamics, 2020_

## Referee Comment (RC1) · Anonymous Referee #1 · 30 Nov 2020

General comments:

Understanding the mechanism of long run-out of a landslide/avalanche is still a great challenge, but plays an important role in correctly describing the landslide dynamics and its deposition morphology. Models and hypotheses have been presented to explain the exceptional run-out of landslide, including fragmentation. The authors mention, that fragmentation may consume energy, potentially at a cost of the runout length. So, they are concerned to specify the integrated effect of fragmentation on the runout dynamics of rock avalanches. By analysing analogue models of dynamically fragmenting rock slides, the authors isolate the effects of friction and fragmentation on rock avalanche

dynamics. For this, they assume that there exists some mechanism that causes a low, but constant effective basal friction. So, the variation in run-out or mobility (called Heim's ratio) is assumed to depend only on fragmentation. This is summarized by presenting a scaling law which shows that the change in the degree of fragmentation can explain the large variation in runout of rock avalanches seen in nature. The authors also compare their experimental results to a set of data (that also utilizes date from some internal report) and discuss their relevance to natural systems. The topic is very important and interesting. The presented mathematical model is one of the main contributions in this paper manuscript that may play a crucial role in describing runout of a landslide. There are some appreciable, clever and novel ideas, and important observations. However, there are also several critical issues on the presented model and other conceptual aspects that must be addressed properly. This mainly concerns the presented mathematical model and explaining the observed results with underlying mechanics. Parameters should be well defined. The paper could have been better organized and discussed.

Specific comments:

Some confusions are already seen in the Abstract: Usually, exceptionally long runout is associated with the large volume that results, e.g., by some fluidization/lubrication effects. This has been discussed by presenting a mechanical analytical model in https://doi.org/10.1016/j.enggeo.2013.01.012. These relevant aspects should have been discussed.

L12: 150 km/h is not that high for rapid avalanche with exceptional run-out.

L17,22: Although friction is assumed to be low and constant in this paper, the above mentioned reference resented the first-ever explicit and unified theoretical model for exceptional mobility of landslide and avalanche: with the consideration of volumetric, physical, and topographical parameters, the authors presented a new model to quantify the scale-dependent friction coefficient of large debris avalanche events. It might be

relevant to discuss.

L27: "additional controlling factors": One such very important, dominant factor is erosion/entrainment that explains the mechanical causes of exceptional long travel distance. This is worth mentioning with reference.

L41-42: "Weassume that there exists some mechanism that causes a low, but constant effective coefficient of basal friction and keep it constant in our model.": This is a clever idea, but is this realistic and observable in nature? Please elaborate with reference.

L58-59: The normalization is a bit strange and not justified! E.g., why the length $L_{spread}$ is normalized by the vertical fall height $H$ and not by other more relevant length scale such as $l_0$? Also, the definition of degree of fragmentation $m_c$ is strange and not discussed why done this way: there can be very few fragmented big boulders and almost all small particles. Then, defining $m_c$ in terms of $m_{max}$ may not be the best representative of the fragmentation. This should be discussed.

L63-71: The readers might ask why these parameter values are chosen.

L74-78: not easy to follow. Not clear which initial conditions are used.

Fig. 2: Figures could be better organized, e.g., by first putting Fig. 3 then Fig. 2; first present model then Fig. 2, etc. The strange behaviors of increasing $H/L$ and $L_{spread}/H$ with large $m_c$ must be clearly discussed. Is this so great to mention about the plotting script in the caption?

L79-82, 85-86: Very interesting/important, novel observation, but the writing should be improved. E.g., does it mean fragmentation results in decreased runout?

L90-92: Appreciable novel observations! However, not quite clear what you really want to say. You have not yet clearly quantified the internal friction and interactions between the fragmented particles. The energy dissipation is also due to loss of momentum because of the early depositions of (many small) fragments. Such reduction in mobility due to deposition has recently been explained with the mechanical erosion model for

mass flows.

L97-118: The following are critical issues that must be properly addressed. There are two main essences of this paper. (i) fragmentation experiments and the analysis of the data, and (ii) developing a mathematical model explaining the runout in terms of fragmentation intensity. I hope the models and the associated figures are right. However, the authors must fix the following: Readers can't follow, please derive equation (3) e.g., in and Appendix. Equation (4) might not be right as it appears now; you need $1/Mg$ in $W$, or? $L_{spread}$ and $W$ are assumed to be implicit functions of frictions and fragmentation, that might be reasonable, but are not quantified. Equations (5), (7) and can't be obtained in a usual way, please check and prove. Further, why do you have mu inside mu? Also, the logarithmic dependency of the work $W$ on fragmentation $m_c$ is not clear, must be discussed. Please check carefully and derive the model equations explicitly, may be in and Appendix. Equations (6), (7): $m_c > 1$, by definition, and also alpha $> 0$, then $- alpha \ln(m_c) < 0$, means $L_{spreading} < 0$? This is not realistic. So, please derive all the equations such that the readers can easily follow and understand the mechanisms behind them.

Technical comments:

The English should be improved (e.g., L2, L14, L41, . . .).

Notations should be clearly defined (e.g., L20, what ap stands for, . . . )

L23: Heim's ratio can be much smaller than 0.1.

Fig. 1: Caption: would be better to replace "measurements" by "scales"?

Fig. 3: Why not the same times for panels on both columns? It is difficult to compare. Also, put scales in x and y axes, and c = ** on top of the columns.

L97, 100: Parameters are not well defined. E.g., what is $L_s$, which length? Please clearly define all parameters and show in the figure.

---

## Referee Comment (RC2) · Anonymous Referee #2 · 2 Mar 2021

General Comments \*\*\*\*\*\*\*\*\*\*\*\*\*\*\*\*\*\*\*\*\*\*\*\*\*\*\*\*\*\*\*\*\*\*\*\*\*\*\*\*\*\*\*\*\*\*\*\*\*\*\*\*\*\*\*\*\*\*\*\*\*\*\*\*\*\*\*\*\*\*\*\*\*\*\*\*\*\*\*\*\*\*
In summary, the present "short communication" (from now on: SC) displays the text-part (introduction, methods, results and discussion, conclusion) of a previous open access publication of experimental data (Haug et al. 2020). Additionally, the present SC constitutes a follow up article to the authors previous publication Haug et al. 2016.

The scientific results and conclusions are presented in a clear, concise, and well-structured way.

The present SC addresses fragmentation, a generally observed feature of rock avalanches, and its role within the still not thoroughly understood emplacement (run-

out) of highly mobile rock avalanches, despite more than a century of research. Fragmentation itself became a field of interest in rock avalanche research, about two decades ago and remained a promising attempt since then. Rock avalanches, because of their size, violent dynamics and nevertheless overall scarce occurrence (luckily), pose a huge challenge in their investigation at laboratory scale or to be addressed within the limits of available computational power, as the authors correctly state in the SC.

The present SC presents and discusses the results of so called 1g laboratory experiments. The drawback of these experiments is, that governing velocities and stress states are reduced by 2 to 3 orders of magnitude compared to the natural prototype model. Furthermore, the model boundaries have to be set so close, that the models resemble not even a millionth of the natural volumetric scale. Other fields of science experience similar problems: Especially physics faces since decades a growing gap between theories proposed by theoretical physics and the (even theoretical) ability of experimental physics to put these theories to observable and repeatable tests which threatens their epistemological validity. Inventing to a certain degree simple but useful experiments for big theoretical questions displays a main quest in physics today. Hence, 1 g experiments (and corresponding, relatively small-scale numerical models) are, what is currently available to many research groups, after reviewing further publications on the current topic. In this sense the present SC matches the, let's call it "relative state of the art", as defined by the cited articles. In that way the SC can be considered a good contribution to scientific progress.

But on the other hand, it shall be stated, that more rigorous publications on the present topic already exist which are more or less ignored by the SC. Especially the publication of Imre et al, 2010 *) not only presents data derived from a physical modelling environment of much higher velocities and stress states, such as much closer to natural situation, but also suggests quantities of energy dissipated by fragmentation. They also suggest what fragmentation due to inter-particle collisions actually may cause –

the dispersive stress model and show, applying a numerical model, how the spreading of a rock avalanche – hence the rapid propagation of the front at minor propagation of the center of mass, emerges from the dispersive stress model. Since velocities and stress states within their physical model applied still not fully resembled the natural model, they present in Imre et al, 2011 **) an brittle analogue material together with a rigorous scaling of all properties of the analogue material according to the physical modelling applied within. Based in these, dependencies are derived how physical properties of natural rock materials, like strength, pre-fracturing etc., governs the run-out of rock avalanches which may became a truly useful tool in practical hazard mitigation in future.

These topics are alle covered or sometimes at least scratched within the SC also, but within Haug et al. 2016, Imre et al., 2010 just serves as yet another reference on fragmentation in rock avalanches without referring to any details. Within the present follow up SC this publication is completely ignored by the authors. To be clear, there is no intention to advocate the work by Imre et al. They applied a complicated and expensive physical modelling environment and it may become useful to refer to 1g experiments instead to speed up research on rock avalanches as I stated above on physics in general. But scientific progress can be achieved only if something which is proposed as a valid attempt, is thoroughly discussed, in its strengths and weaknesses, to its predecessor, or in the case of Imre et al., 2010, to a model which is by physics much closer to the natural situation than the 1g experiment presented within the SC. Since the present SC lacks such a proper discussion, the SC is considered just a fair overall contribution to scientific progress, although the scaling law presented is innovative. The fact that currently a number of research groups apply 1g experiments of the kind presented within the SC, proofs by its own their popularity but not necessarily their suitability for lasting progress in rock avalanche research.

*) Imre, B., J. Laue, and S. M. Springman (2010), Fractal fragmentation of rocks within sturzstroms: Insight derived from physical experiments within the ETH geotechnical

drum centrifuge, Granular Matter, 12(3), 267–285, doi:10.1007/s10035-009-0163-1 **)
Imre, B., Wildhaber, B., and S. M. Springman (2011), A physical analogue material to simulate sturzstroms. Int. J. of Physical Modelling in Geotechnics 2011 11:2, 69-86.

Specific Comments ***********************************************************************************
In section 3.3 the authors apply their experimentally derived data to "a natural data set". This data set resembles nine rock avalanches reported by Locat et al. (2006). Due to comprehensible reasons the authors derive a data fit to four natural cases. According to the text theses four cases "cover a range of two orders of magnitude (from 2 Âů 10*6 to 90 Âů 10*6 m3)".

Based on these fit authors claim:

"The similarity seen between experimental and natural data suggests universality with respect to the empirical constants and that the rock avalanches considered here all have a close to constant effective friction of about 0.15."

"This shows that fragmentation plays a governing role in the runout of rock avalanches and should be included in hazard assessments."

In section 4 the author finally come the conclusion that: "The law is validated against a natural data set proving its universality and predictive power."

First: A data set of four fits out of nine cases is extremely limited. There are data of much more cases of rock avalanches available. Furthermore, known cases of rock avalanches range from 1 10*6 to 1 10*10 m3, these are 4 orders of magnitude. Adding Martian rock avalanches this range extends to 7 orders of magnitude. Therefore, the data presented within the SC are in fact limited to a very narrow fit giving no justification for the claimed "universality" and "predictive power" of their derived fit. These terms shall be omitted therefore as unproven.

Second: The fact that the experiments yielded an effective friction of about 0.15 displays an interesting observation, but for this class of rock avalanches this was known

**ESurfD**
since Albert Heim. It remains unclear how fragmentation contributes to such a low friction within the experiment, hence how it "plays a governing role in the runout of rock avalanches", especially since fragmentation has been identified as a major energy sink at the same time. Therefore, while fragmentation truly displays an important role in further research on rock avalanches, and this SC contributes to it, this SC provides no hint at all, how a fragmentation shall be "included in hazard assessments" in practise. This claim shall be omitted.

Technical Corrections **********************************************************************************
None.

---

## Editor Comment (EC1) · Michael Krautblatter (Editor) · 23 Mar 2021

Dear Dr. Haug and coauthors,

we have now received two detailed reviews of your paper "Short communication: Runout of rock avalanches limited by basal friction but controlled by fragmentation" from well-established experts in this field. Both highlight the importance and novelty of your paper. Reviewer 2 asks for minor revision but suggests a number of improvements regarding the discussion of scaling issues and prior work and provides detailed comments on a number of your statements in the specific comments. Reviewer 1 suggests

major revisions and criticizes aspects of the presented mathematical model, the parameterization, the organization and discussion in the paper. In addition, a significant number of specific comments provide suggestions to improve the assumptions, the model, the parameterization (C2-C3) and the discussion (C4) of your paper. I suggest to carefully address the comments made by both reviews and to implement improvements in your paper as part of a moderate revision. Reviewer 1 indicated that he/she is willing to review the revised version of manuscript after the implementation of the changes.

All the best

Michael Krautblatter

---

## Author Comment (AC1) · 25 Mar 2021

Reviewer comments in black

Author response in blue

*"text changes"*

**Anonymous Referee #1**

General comments:

Understanding the mechanism of long run-out of a landslide/avalanche is still a great challenge, but plays an important role in correctly describing the landslide dynamics and its deposition morphology. Models and hypotheses have been presented to explain the exceptional run-out of landslide, including fragmentation. The authors mention, that fragmentation may consume energy, potentially at a cost of the runout length. So, they are concerned to specify the integrated effect of fragmentation on the runout dynamics of rock avalanches. By analysing analogue models of dynamically fragmenting rock slides, the authors isolate the effects of friction and fragmentation on rock avalanche. For this, they assume that there exists some mechanism that causes a low, but constant effective basal friction. So, the variation in run-out or mobility (called Heim's ratio) is assumed to depend only on fragmentation. This is summarized by presenting a scaling law which shows that the change in the degree of fragmentation can explain the large variation in runout of rock avalanches seen in nature. The authors also compare their experimental results to a set of data (that also utilizes date from some internal report) and discuss their relevance to natural systems.

The topic is very important and interesting. The presented mathematical model is one of the main contributions in this paper manuscript that may play a crucial role in describing runout of a landslide. There are some appreciable, clever and novel ideas, and important observations.

However, there are also several critical issues on the presented model and other conceptual aspects that must be addressed properly. This mainly concerns the presented mathematical model and explaining the observed results with underlying mechanics. Parameters should be well defined. The paper could have been better organized and discussed.

We thank the reviewer for this positive view. We revised the math section for clarity and discussed the critical points raised now more specifically.

Specific comments:

Some confusions are already seen in the Abstract: Usually, exceptionally long runout is associated with the large volume that results, e.g., by some fluidization/lubrication effects. This has been discussed by presenting a mechanical analytical model in https://doi.org/10.1016/j.enggeo.2013.01.012. These relevant aspects should have been discussed.

We agree that weakening processes involving fluids are of prime importance. However, the experiments aimed at isolating the effect of fragmentation and having all other parameters as constant as possible. Amongst those constant parameters in the experiment is the friction coefficient which Pudasaini and Miller (2013) argue is scale-dependent and controlled by

fluidization, i.e. pore fluid pressure. Accordingly, this is especially significant in submarine environments while in dry subaerial, terrestrial, non-volcanic environments that we focus on the friction coefficient seems less affected by fluidization. For example, looking at the data shown in Pudasaini and Miller (2013) for "non-volcanic events" it seems this subset (which is the one relevant for us) shows the smallest range in friction coefficient and least sensitivity to volume compared to others. The data and model have a plateau in the volume midrange suggesting insensitivity for at least some part of the spectrum. At the same time, it is this part that shows the greatest variability for a given volume in the whole data set (suggesting additional, likely similar important controlling factors). So, while we appreciate the validity of their model over a wide range of sizes and environments, it seems that at least in those environments that we focus on, additional controlling factors are at least equally important.

We discuss this limitation/focus of our study now more specifically in the introduction:

*"One of the best established, but perhaps least understood observations of rock avalanches, is the dependence of the Heim's ratio on volume: rockslides below a size of approximately 10$^6$\,m$^3$ all have a relatively constant Heim's ratio of $\sim$0.4-0.7, but for larger rockslides it decreases with volume, reaching values $<$0.1 for volumes larger than 10$^9$ m$^3$ \citep{Pudasaini2013,Lucas2014}. This suggests a scale-dependent mechanism of decreasing apparent friction with volume that becomes dominant at large volumes \citep{Davies1999runout}. Analytical modelling and numerical simulation involving lubrication mechanisms by \cite{Pudasaini2013} and \citet{Lucas2014} provided mechanical explanations for this observation. Importantly, however, even within a narrow range of volumes, runouts are seen to span orders of magnitude suggesting additional controlling factors on runout that are insensitive to size."*

L12: 150 km/h is not that high for rapid avalanche with exceptional run-out.

We found it difficult to constrain since direct observations have been rare. We include now also the 100m/s used as an upper bound for the modelling in Legros (2002):

*"With volumes larger than $10^6$\,m$^3$, and speeds reported at over 150\,km/h \citep{Campbell1989} and possibly up to 100m/s \citep{Legros2002}, the destructive power of rock avalanches is unprecedented."*

L17,22: Although friction is assumed to be low and constant in this paper, the above mentioned reference resented the first-ever explicit and unified theoretical model for exceptional mobility of landslide and avalanche: with the consideration of volumetric, physical, and topographical parameters, the authors presented a new model to quantify the scale-dependent friction coefficient of large debris avalanche events. It might be relevant to discuss.

We mention this model now more specifically (see reply above). However, we here focus on the scale-independent effects seen (for example in the data set used by Pudasaini and Miller (2013):

*"Importantly, however, even within a narrow range of volumes, runouts are seen to span orders of magnitude suggesting additional controlling factors on runout that are insensitive to size."*

we further specify this limited scope:

*"To isolate the scale-independent effect of fragmentation we keep both the volume and friction within a narrow range in our models compared to nature. Note, this approach explicitly excludes dynamic weakening mechanisms that are suspected in natural prototypes. Specifically, our models do not include fluids and frictional heating is insignificant such that lubrication mechanisms \citep[e.g.][]{Pudasaini2013,Lucas2014} do not play a role. Granular pressurization \citep[e.g.][]{Imre2010} is also not considered significant in our experiments because of the low energy involved. Other potentially important mechanisms like bedrock erosion \citep[e.g.][]{Hungr2004,Pudasaini2020} are excluded here for simplicity. The experimental design, therefore, means that the observed variation in Heim's ratio is due to fragmentation and dry friction."*

*"...our results suggest that the variation seen in Heim's ratio for these rock avalanches are not (only) caused by scale-dependent basal friction, but by differing degrees of fragmentation."*

L27: "additional controlling factors": One such very important, dominant factor is erosion/entrainment that explains the mechanical causes of exceptional long travel distance. This is worth mentioning with reference.

We agree that erosion is an important mechanism lowering apparent basal friction. We now included it with reference to https://doi.org/10.1130/B25362.1 and https://doi.org/10.1016/j.ijmultiphaseflow.2020.103416 in the first paragraph.

*"This suggests low effective basal frictions $\mu_{eff}$, which is usually attributed to various dynamic weakening processes \citep[e.g.][]{Kent1966,Shreve1968,HSU1975,Melosh1979,Campbell1989,Pudasaini2013,Legros2002,Lucas2014,Wang2017} or additional basal erosion processes \citep[e.g.][]{Hungr2004,Pudasaini2020}."*

However, our models do not include basdal erosion so we added it to the list of limitations/simplifications of our models:

*"... Specifically, our models do not include fluids and frictional heating is insignificant such that lubrication mechanisms \citep[e.g.][]{Pudasaini2013,Lucas2014} do not play a role. Granular pressurization \citep[e.g.][]{Imre2010} is also considered not significant in our experiments because of the low energy involved. Other potentially important mechanisms like bedrock erosion \citep[e.g.][]{Hungr2004,Pudasaini2020} are excluded here for simplicity."*

L41-42: "We assume that there exists some mechanism that causes a low, but constant effective coefficient of basal friction and keep it constant in our model.": This is a clever idea, but is this realistic and observable in nature? Please elaborate with reference.

A constant friction coefficient at natural scale is likely not realistic while in the lab it is (we tested the rate and state dependency of sands used here and found no significant weakening). Since we reduce natural complexity in our analogue model this assumption is part of our strategy to isolate the fragmentation effect. In summary of the above comments and replies, we would like to keep the list of potential weakening mechanisms short as our models do not include most and are intentionally simplified and necessarily limited. We specify our approach:

*"To isolate the scale-independent effect of fragmentation we keep both the volume and friction within a narrow range in our models compared to nature. Note, this approach explicitly excludes dynamic weakening mechanisms that are suspected in natural prototypes. Specifically, our models do not include fluids and frictional heating is insignificant such that lubrication mechanisms \citep[e.g.][]{Pudasaini2013,Lucas2014} do not play a role. Granular pressurization \citep[e.g.][]{Imre2010} is also not considered significant in our experiments because of the low energy involved. Other potentially important mechanisms like bedrock erosion \citep[e.g.][]{Hungr2004,Pudasaini2020} are excluded here for simplicity. The experimental design, therefore, means that the observed variation in Heim's ratio is due to fragmentation and dry friction."*

L58-59: The normalization is a bit strange and not justified! E.g., why the length L_spread is normalized by the vertical fall height H and not by other more relevant length scale such as l_0?

We agree that L_spread/l_0 is an intuitive ratio and describes the extension of the slide. On the other hand, L_spread is similarly correlated to H as L is and so both are at first order describing the conversion of potential energy into translation and deformation, respectively. Moreover, this ratio drops out of our scaling law (eq. 5) suggesting it has a physical meaning beyond the purely geometric meaning of L_spread/l_0 (extension).

We specified this choice now:

*"We normalize $L_{spread}$ by fall height $H$ to have a parameter describing the conversion of potential energy into spreading equivalent to Heim's ratio."*

Also, the definition of degree of fragmentation m_c is strange and not discussed why done this way: there can be very few fragmented big boulders and almost all small particles. Then, defining m_c in terms of m_max may not be the best representative of the fragmentation. This should be discussed.

It is true that in models we could derive statistically more quantitative parameters describing the amount of fragmentation, e.g. based on the full fragment size distribution. However, we here explicitly aim at a proxy for the fragmented volume which is accessible and easy to derive in nature, too. It is therefore a trade-off between capturing the process accurately in models and accessibility of the information in nature (and from literature). Concerning energy consumption due to fragmentation we feel it is intuitive that the largest fragment, which remains intact and controls the complementary fragmented volume, is a valid first-order proxy for the energy consumed by fragmentation. In Haug et al. (2016), we verified the usefulness of m_c by benchmarking it against the breakage parameter used in previous studies (e.g., Bowman et al., 2012; Langlois et al., 2015).

We clarify this by saying:

*"We choose this rather simple parameter, which has been validated and benchmarked against breakage parameters used by previous studies in \citet{Haug2016}, as a tradeoff between capturing the process accurately in models and accessibility of the information in nature."*

L63-71: The readers might ask why these parameter values are chosen.

These were experimental constraints. In analogue modelling, we are limited in the parameter space and try to extend it as much as possible with the materials we have which may result in odd numbers. For clarification we added:

*"Combining these sets of data from various experiments allows for covering a wide enough parameter space for the analysis in this study."*

L74-78: not easy to follow. Not clear which initial conditions are used.

We rephrase:

*"To quantitatively analyse the experiments we focus on the correlation between runout and fragmentation and neglect all other parameters. This is justified by the collapse of experimental and natural data in Figure~\ref{fig:f03}a: When plotting the Heim's ratio against fragmentation ($m_c$), all data collapse to the same trend and therefore no distinction is made between the experimental data in this figure. Qualitatively, Heim's ratio decreases rapidly for low to intermediate degrees of fragmentation, reaching a minimum at $m_c \approx 5$ of about 0.2 and increases again slightly for higher degrees of fragmentation. A similar relation is observed between the length of the deposits (Figure~\ref{fig:f03}b), which increases with fragmentation until $m_c \approx 5$ and slightly decreases beyond."*

Fig. 2: Figures could be better organized, e.g., by first putting Fig. 3 then Fig. 2; first present model then Fig. 2, etc.

OK, we followed this suggestion and switched figure 2 and 3.

The strange behaviors of increasing H/L and L_spread/H with large m_c must be clearly discussed.

The trends seen in these two plots of now Fig 3 are clearly correlated and suggest an intrinsic relation between spreading and runout: Below m_c=5 the increase in L_spread causes H/L to decrease while above that threshold the spreading decreases and H/L increases consistently suggesting energy consumption.

We specified in the figure caption:

*"Note the opposite trends of the two curves suggesting an intrinsic relationship between spreading and runout."*

We hope the shape and relation of these two curves become clearer with the revised structure and wording in section 3.1.

Is this so great to mention about the plotting script in the caption?

OK, we omitted this.

L79-82, 85-86: Very interesting/important, novel observation, but the writing should be improved. E.g., does it mean fragmentation results in decreased runout?

For larger degrees of fragmentation this is correct (m_c>5) because fragmentation consumes energy (Haug et al. 2016 and references therein). Below m_c=5 fragmentation seems to increase mobility and runout.

We clarify now and also include deposition:

*"The stronger sample (Figure~\ref{fig:f02}A) is observed to fragment less than the weaker one (Figure~\ref{fig:f02}B). Thereafter, fragments of the stronger sample spread with limited interaction while the fragments from the weaker sample collide and/or slide next to each other and deposition starts relatively early. We infer, at first order, that while mobility generally increases with fragmentation, a higher amount of internal deformation is experienced along with increased fragmentation and increased deposition."*

L90-92: Appreciable novel observations! However, not quite clear what you really want to say. You have not yet clearly quantified the internal friction and interactions between the fragmented particles.

The fragmentation effect on runout for larger fragmentation degrees (energy sink) has been described in more detail in Haug et al. 2016 and we here take up their findings. A more quantitative analysis follows in section 3.2.

We have re-phrased that part "preparing" the quantitative section for clarity also including the effect of deposition and loss of momentum:

*"Considering the increased internal deformation observed with the degree of fragmentation (Figure~\ref{fig:f02}), the reduction of runout with $m\_c>5$ appears to be the result of the increased energy dissipation through internal friction within the rock mass as well as an increase in basal friction as the sliding surface becomes rougher due to syn-sliding deposition \citep[e.g.][]{Pudasaini2020}. A loss of mass and therefore momentum due to deposition may additionally result in deceleration and reduced runout as a function of $m\_c$ \citep[e.g.][]{Pudasaini2020}. Consequently, the minimum of the Heim's ratio observed in Figure~\ref{fig:f03}a appears as the result of a competition between spreading enhancing mobility and the energy-consuming fragmentation process."*

The energy dissipation is also due to loss of momentum because of the early depositions of (many small) fragments. Such a reduction in mobility due to deposition has recently been explained with the mechanical erosion model for mass flows.

That's a good point. Thank you for emphasizing this. We included this now specifically:

*"Considering the increased internal deformation observed with the degree of fragmentation (Figure~\ref{fig:f02}), the reduction of runout with $m\_c>5$ appears to be the result of the increased energy dissipation through internal friction within the rock mass as well as an increase in basal friction as the sliding surface becomes rougher due to syn-sliding deposition \citep[e.g.][]{Pudasaini2020}. A loss of mass and therefore momentum due to deposition may additionally result in deceleration and reduced runout as a function of $m\_c$ \citep[e.g.][]{Pudasaini2020}."*

L97-118: The following are critical issues that must be properly addressed. There are two main essences of this paper. (i) fragmentation experiments and the analysis of the data, and

(ii) developing a mathematical model explaining the runout in terms of fragmentation intensity. I hope the models and the associated figures are right. However, the authors must fix the following:

Readers can't follow, please derive equation (3) e.g., in and Appendix.

We added a new figure A01 in the appendix where all the distances used to derive eq. 3 are defined.

Equation (4) might not be right as it appears now; you need 1/Mg in W, or?

That is true, thanks for pointing us to this error.

L_spread and W are assumed to be implicit functions of frictions and fragmentation, that might be reasonable, but are not quantified.

True, these are reasonable assumptions that we do not quantify. Likely a dedicated study would be needed but this would be beyond the scope of this "Short communication".

Equations (5), (7) and can't be obtained in a usual way, please check and prove.

We checked the math and corrected it where necessary. Eq. (5) follows in a step of rearrangement from eq. (4) while the derivation of eq. (7) (now eq. (8)!) is now presented in one step more as resulting from inserting (6) and (7) into (5). We hope the revised math section is clearer now.

Further, why do you have mu inside mu?

We think there is a misunderstanding: The bracket indicates a factor term, not a function variable. Mu appears at two positions in that product.

Also, the logarithmic dependency of the work W on fragmentation m_c is not clear, must be discussed.

This is an inference based on the trend shown in Fig. 8b in Haug et al. (2016). We now specified and justified it better:

*"...the experimental work by \citet{Haug2016} suggests that dissipative energy loss through fragmentation increases less for higher degrees of fragmentation and therefore can be described with a logarithmic function of $m_c$:"*

Please check carefully and derive the model equations explicitly, may be in and Appendix. Equations (6), (7): m_c > 1, by definition, and also alpha > 0, then – alpha ln(m_c) < 0, means L_spreading <0? This is not realistic.

We realize that the order of terms on both sides of the formula was switched which is mathematically not relevant but from which confusion may have arisen. We now treat and explain the two terms separately and correct the signs.

So, please derive all the equations such that the readers can easily follow and understand the mechanisms behind them.

We revised this section for clarity, corrected minor errors and decluttered the math. We hope it is clearer now.

Technical comments:

The English should be improved (e.g., L2, L14, L41, . . .).

*OK, the revised version finally has been checked by Jon Bedford (native speaker).*

Notations should be clearly defined (e.g., L20, what ap stands for, . . . )

*True, that was not clear. "ap" stands for "apparent", we clarified this now:*

*"The resulting ratio*
*\begin{linenomath*}*
*\begin{equation}*
*   \mu_{apparent} = \frac{H}{L}*
*\label{eq:Heim}*
*\end{equation}*
*\end{linenomath*}*
*is known as the Heim's ratio \citep[as cited in \citealp{HSU1975}]{heim1882bergsturz} and serves as a proxy for $\mu_{eff}$ when called the "apparent" coefficient of friction \citep{Manzella2012}."*

L23: Heim's ratio can be much smaller than 0.1.

*True, but for those terrestrial, non-volcanic cases we focus on 0.1 appears as a lower bound (e.g. https://doi.org/10.1038/ncomms4417, https://doi.org/10.1016/j.enggeo.2013.01.012)*

Fig. 1: Caption: would be better to replace "measurements" by "scales"?

*OK, we modified:*

*"sketch of the slope geometry of experiments, relevant parameters and length scales..."*

Fig. 3: Why not the same times for panels on both columns? It is difficult to compare.

*The time indicates the time since the first impact while we have chosen increments of equal travel distance. We explain this more specifically now:*

*"\caption{\textbf{Snapshots from the experiments}: (a) intermediate strength sample ($C = 40$\,kPa) and (b) low strength sample ($C = 4$\,kPa).The red lines in the upper images indicate the geometry of the basal plates. Images are chosen to represent similar travel distances in (a) and (b). The time given above each image reflects the time since the first impact. ..."*

Also, put scales in x and y axes, and c = ** on top of the columns.

*We added a scale bar and indicated the cohesion as suggested.*

L97, 100: Parameters are not well defined. E.g., what is L_s, which length? Please

clearly define all parameters and show in the figure.

*L_s is the length of the slope. We specified it in the text and in a new figure A01 which illustrates how the equation is derived.*

---

## Author Comment (AC2) · 25 Mar 2021

Reviewer comments in black

Author response in blue

*"text changes"*

**Anonymous Referee #2**

General Comments

In summary, the present "short communication" (from now on: SC) displays the textpart (introduction, methods, results and discussion, conclusion) of a previous open access publication of experimental data (Haug et al. 2020). Additionally, the present SC constitutes a follow up article to the authors previous publication Haug et al. 2016. The scientific results and conclusions are presented in a clear, concise, and well structured way.

The present SC addresses fragmentation, a generally observed feature of rock avalanches, and its role within the still not thoroughly understood emplacement of highly mobile rock avalanches, despite more than a century of research. Fragmentation itself became a field of interest in rock avalanche research, about two decades ago and remained a promising attempt since then. Rock avalanches, because of their size, violent dynamics and nevertheless overall scarce occurrence (luckily), pose a huge challenge in their investigation at laboratory scale or to be addressed within the limits of available computational power, as the authors correctly state in the SC.

The present SC presents and discusses the results of so called 1g laboratory experiments. The drawback of these experiments is, that governing velocities and stress states are reduced by 2 to 3 orders of magnitude compared to the natural prototype model. Furthermore, the model boundaries have to be set so close, that the models resemble not even a millionth of the natural volumetric scale. Other fields of science experience similar problems: Especially physics faces since decades a growing gap between theories proposed by theoretical physics and the (even theoretical) ability of experimental physics to put these theories to observable and repeatable tests which threatens their epistemological validity. Inventing to a certain degree simple but useful experiments for big theoretical questions displays a main quest in physics today. Hence, 1 g experiments (and corresponding, relatively small-scale numerical models) are, what is currently available to many research groups, after reviewing further publications on the current topic. In this sense the present SC matches the, let's call it "relative state of the art", as defined by the cited articles. In that way the SC can be considered a good contribution to scientific progress.

But on the other hand, it shall be stated, that more rigorous publications on the present topic already exist which are more or less ignored by the SC. Especially the publication of Imre et al, 2010 *) not only presents data derived from a physical modelling environment of much higher velocities and stress states, such as much closer to natural situation, but also suggests quantities of energy dissipated by fragmentation. They also suggest what fragmentation due to inter-particle collisions actually may cause – C2 ESurfD Interactive comment Printer-friendly version Discussion paper the dispersive stress model and show, applying a numerical model, how the spreading of a rock avalanche – hence the rapid propagation of the front at minor propagation of the center of mass, emerges from the dispersive stress model. Since velocities and stress states within their physical model applied still not fully resembled the natural model, they present in Imre et al, 2011 **) an brittle analogue material together with a rigorous scaling of all properties of the analogue

material according to the physical modelling applied within. Based in these, dependencies are derived how physical properties of natural rock materials, like strength, pre-fracturing etc., governs the run-out of rock avalanches which may became a truly useful tool in practical hazard mitigation in future.

These topics are alle covered or sometimes at least scratched within the SC also, but within Haug et al. 2016, Imre et al., 2010 just serves as yet another reference on fragmentation in rock avalanches without referring to any details. Within the present follow up SC this publication is completely ignored by the authors. To be clear, there is no intention to advocate the work by Imre et al. They applied a complicated and expensive physical modelling environment and it may become useful to refer to 1g experiments instead to speed up research on rock avalanches as I stated above on physics in general. But scientific progress can be achieved only if something which is proposed as a valid attempt, is thoroughly discussed, in its strengths and weaknesses, to its predecessor, or in the case of Imre et al., 2010, to a model which is by physics much closer to the natural situation than the 1g experiment presented within the SC.

Since the present SC lacks such a proper discussion, the SC is considered just a fair overall contribution to scientific progress, although the scaling law presented is innovative. The fact that currently a number of research groups apply 1g experiments of the kind presented within the SC, proofs by its own their popularity but not necessarily their suitability for lasting progress in rock avalanche research.

*) Imre, B., J. Laue, and S. M. Springman (2010), Fractal fragmentation of rocks within sturzstroms: Insight derived from physical experiments within the ETH geotechnical C3 ESurfD Interactive comment Printer-friendly version Discussion paper drum centrifuge, Granular Matter, 12(3), 267–285, doi:10.1007/s10035-009-0163-1

**) Imre, B., Wildhaber, B., and S. M. Springman (2011), A physical analogue material to simulate sturzstroms. Int. J. of Physical Modelling in Geotechnics 2011 11:2, 69-86.

We thank the reviewer for this review sharing his view from a geotechnical perspective. We agree that experiments using real rock material (or closer to real rock like is "ETHAR") under conditions closer to the prototype instead of weaker rock analogue material under downscaled laboratory conditions provide highly valuable and unique data. This is especially true for those cost-intensive experiments run in a centrifuge. However, an "analogue modelling" approach as used here should then be considered a different class of experiments equivalent to the distinction made in the geoscience community between rock mechanics experiments (e.g. using High pressure-high temperature deformation devices) and any deformation experiments using weaker (kPa instead MPa) analogue material under consistently (according to scaling laws) lower P-T conditions. Despite this downscaling issue, analogue models have some advantages beyond being cheap: They are also more accessible in terms of monitoring the deformation process more directly at high spatial and especially temporal resolution typically not realizable in geotechnical experiments using real rocks under close to real conditions. In the end, we think both, experiments using quasi-real rock material and those with weak analogue rock material, should be considered complementary with each of them having its weight when contributing to the discussion. We appreciate the work by Imre and others using the geotechnical approach as we do appreciate the many numerical studies. However, within the scope of this short communication, we feel it is impossible to discuss and we necessarily limit our scope. Nevertheless, we feel that to embrace the wider community we now more explicitly acknowledge the limitation of our approach including the lack of various processes including the "granular pressure" effects described by Imre et al.:

*""To isolate the scale-independent effect of fragmentation we keep both the volume and friction within a narrow range in our models compared to nature. Note, this approach explicitly excludes dynamic weakening mechanisms that are suspected in natural prototypes. Specifically, our models do not include fluids and frictional heating is insignificant such that lubrication mechanisms \citep[e.g.][]{Pudasaini2013,Lucas2014} do not play a role. Granular pressurization \citep[e.g.][]{Imre2010} is also not considered significant in our experiments because of the low energy involved. Other potentially important mechanisms like bedrock erosion \citep[e.g.][]{Hungr2004,Pudasaini2020} are excluded here for simplicity. The experimental design, therefore, means that the observed variation in Heim's ratio is due to fragmentation and dry friction."*

Specific Comments

In section 3.3 the authors apply their experimentally derived data to "a natural data set". This data set resembles nine rock avalanches reported by Locat et al. (2006). Due to comprehensible reasons the authors derive a data fit to four natural cases. According to the text theses four cases "cover a range of two orders of magnitude (from 2 Â ˚u 10*6 to 90 Â ˚u 10*6 m3)". Based on these fit authors claim: "The similarity seen between experimental and natural data suggests universality with respect to the empirical constants and that the rock avalanches considered here all have a close to constant effective friction of about 0.15." "This shows that fragmentation plays a governing role in the runout of rock avalanches and should be included in hazard assessments." In section 4 the author finally come the conclusion that: "The law is validated against a natural data set proving its universality and predictive power."

First: A data set of four fits out of nine cases is extremely limited. There are data of much more cases of rock avalanches available. Furthermore, known cases of rock avalanches range from 1 10*6 to 1 10*10 m3, these are 4 orders of magnitude. Adding Martian rock avalanches this range extends to 7 orders of magnitude. Therefore, the data presented within the SC are in fact limited to a very narrow fit giving no justification for the claimed "universality" and "predictive power" of their derived fit. These terms shall be omitted therefore as unproven.

Indeed, volume was not a variable in our model (intentionally) and we do not contribute to the discussion about scale dependency of Heim's ratio. Instead, we focus here on the about one order of magnitude variability in Heim's ratio seen for similar-sized (!) avalanches. This reduces both the "universality" of our result (we agree!) but also the natural data available which is a subset of the arguably larger data set if all volumes (and environments) are considered. Note also, we focus on dry slides and need to extract information about the fragmentation from the original reports. This further narrows down the number of examples we can use. We finally decided to rely on a single source of natural cases (i.e. Locat et al., 2006) because these data are harmonized.

From that small data set, we use most (actually 6) out of nine cases for fitting and provide  reasoning why the remaining 3 cannot be used. They plot consistent with the individual late-stage processes leading us to discard them above the others. One may try to correct for those effects, however, this would be only qualitative, so we omitted this.

We added a note:

*"Note that in all three discarded cases, the late-stage processes tend to increase the expected Heim's ratio and they consistently plot above the trend of the other data in Figure 3b."*

and de-emphasized the universality and use a less provocative terminology, e.g.:

*"...shows that variation in the degree of fragmentation can contribute to the large variation in runout of rock avalanches seen in nature." (in Abstract)*

*"To study the effects of friction and fragmentation on rock avalanche dynamics, we here analyse analogue models of dynamically fragmenting rock slides of similar size." (in Introduction)*

*"The agreement between these slide deposit lengths and the extrapolation of the experimental trend through Equation~\ref{eq:Heim_3} (Figure~\ref{fig:f03}B) supports the validity {deleted: and predictive power} of our proposed scaling law." (in application)*

*"The similarity seen between experimental and natural data suggests some universality concerning the empirical constants." (in application)*

*"The scaling law is validated against a natural data set verifying its applicability." (in Conclusions)*

With respect to scale dependency of runout we specify:

*"...our results suggest that the variation seen in Heim's ratio for these rock avalanches are not (only) caused by scale-dependent basal friction, but by differing degrees of fragmentation."*

*"The scaling law approaches an extreme for which runout is maximized and limited only by basal friction, which itself might be volume-dependent as suggested by earlier studies."*

Second: The fact that the experiments yielded an effective friction of about 0.15 displays an interesting observation, but for this class of rock avalanches this was known since Albert Heim.

We agree the effective friction is close to the basal friction implemented which in turn is similar to natural examples. We specify:

*"We use silicate glass as our substrate, on which the basal friction coefficient is ca. 0.15-0.20 \citep{Haug2016} similar to lowermost values found in natural prototypes \citep{Pudasaini2013,Lucas2014}."*

The interesting point for us was, however, that the experiments yield effective frictions higher than the prescribed one which we attribute to fragmentation.

It remains unclear how fragmentation contributes to such a low friction within the experiment, hence how it "plays a governing role in the runout of rock avalanches", especially since fragmentation has been identified as a major energy sink at the same time. Therefore, while fragmentation truly displays an important role in further research on rock avalanches, and this SC contributes to it, this SC provides no hint at all, how a fragmentation shall be "included in hazard assessments" in practise. This claim shall be omitted.

We feel there is a misunderstanding here. The low friction coefficient (0.15-0.20) is prescribed in the model when using cemented sand sliding on a glass surface. We verified this using friction tests. The observation that we focus on here is not that friction can be so low but rather that apparent friction (Heim's ratio) is larger than that for most of the fragmentation spectrum. So fragmentation is not meant to lower the apparent friction but to increase it, which is in line with the view of fragmentation as an energy sink.

Technical Corrections **************************************************************************
None. Interactive comment on Earth Surf. Dynam.

---

## Author Comment (AC3) · 25 Mar 2021

Dear Prof. Krautblatter,

Thank you for first of all for making it possible to get the reviews from two distinguished colleagues. We know how difficult it is to find reviewers for this rather specific topic and method, especially in these times. We thoroughly revised the manuscript according to their very valuable comments. In particular, we acknowledged now more explicitly the limitation of our analogue modeling approach also concerning previous studies mentioned by both reviewers. We revised the presentation of the mathematical model
for clarity and added one figure in the appendix in that framework. Both reviewers stressed implicitly the volume dependency of runout and pointed to previous models explaining it that should be better discussed along with scale-dependent processes dynamically lowering basal friction. However, since we explicitly focus on the scale-independent aspects of variation in runout and keep the volume and basal friction as a fixed parameter, we did not discuss those models in-depth but rather included the respective processes with references in the description of limitations and scope of our models. We hope it is clearer now how our study contributes to that discussion. In that context, we de-emphasized the "universality" of our scaling law. Finally, we let a native speaking colleague proofread the revised version.

Best regards,

Matthias Rosenau on behalf of all co-authors